# The PIKfyve Inhibitor Apilimod: A Double-Edged Sword against COVID-19

**DOI:** 10.3390/cells10010030

**Published:** 2020-12-27

**Authors:** Maksim V. Baranov, Frans Bianchi, Geert van den Bogaart

**Affiliations:** 1Department of Molecular Immunology and Microbiology, Groningen Biomolecular Sciences and Biotechnology Institute, University of Groningen, 9747AG Groningen, The Netherlands; m.baranov@rug.nl (M.V.B.); f.bianchi@rug.nl (F.B.); 2Department of Tumor Immunology, Radboud Institute for Molecular Life Sciences, Radboud University Medical Center, 6525GA Nijmegen, The Netherlands

**Keywords:** SARS-CoV-2, COVID-19, apilimod, LAM-002A, STA-5326, PIKfyve

## Abstract

The PIKfyve inhibitor apilimod is currently undergoing clinical trials for treatment of COVID-19. However, although apilimod might prevent viral invasion by inhibiting host cell proteases, the same proteases are critical for antigen presentation leading to T cell activation and there is good evidence from both in vitro studies and the clinic that apilimod blocks antiviral immune responses. We therefore warn that the immunosuppression observed in many COVID-19 patients might be aggravated by apilimod.

## 1. Apilimod as Drug Candidate for Treatment and Prevention of COVID-19

In a screen of 12,000 clinical-stage or FDA-approved small molecules by Riva et al. [1], apilimod (also known as LAM-002A or STA-5326) was identified as the most potent drug for blocking replication of SARS-CoV-2 in iPSC-derived pneumocyte-like cells. Apilimod was also found to block entry of SARS-CoV-2 pseudovirus in other cell lines [2]. Apilimod blocks trafficking between lysosomes and endosomes and the trans-Golgi network by inhibiting the cytosolic 5-phosphoinositide kinase PIKfyve [3,4], which results in “swollen” endocytic vacuoles and somehow prevents SARS-CoV-2 invasion [5]. In June 2020, AI Therapeutics, Inc. launched clinical trials to evaluate the treatment efficacy of apilimod in adults with a confirmed SARS-CoV-2 infection (NCT04446377; currently in phase 2).

## 2. How Can Apilimod Prevent Host Cell Invasion of SARS-CoV-2?

We recently showed that apilimod inhibits the cathepsin class of lysosomal proteases [6] and now argue that this underlies its antiviral effects. Following its binding to the ACE2 receptor on the surface of host cells, the spike protein S of SARS-CoV-2, required for fusion of the viral capsid with the host membrane, needs to be proteolytically activated by host cell proteases (Figure 1A) [2]. Depending on the cell type, different host cell proteases can be involved, especially furin [7], TMPRSS2 (transmembrane serine protease 2) [2], but also other proteases (PC1, trypsin, matriptase, cathepsin B/S/L) (Figure 1A) [8]. Indeed, all other drugs identified in the screen by Riva et al. were inhibitors of cysteine proteases [1]. It thus seems likely that the protease inhibiting effect of apilimod interferes with SARS-CoV-2 invasion. However, in contrast to other members of the Coronaviridae, MERS-CoV and SARS-CoV, which invade host cells predominantly via the lumen of endosomes, SARS-CoV-2 mainly invades at the plasma membrane [9,10]. It is therefore unclear how inhibition of lysosomal cathepsins can block viral invasion at the plasma membrane.

## 3. Does Apilimod Prevent SARS-CoV-2 Invasion by Inhibition of Activation of Proteases?

Because uncontrolled proteolytic activity can be harmful for cells, activation of proteases is tightly controlled by proteolytic activation in the trans-Golgi network and in post-Golgi compartments of endo/lysosomal nature [11,12]. For example, for the activation of newly synthesized furin, an autoinhibitory fragment needs to be proteolytically removed in the trans-Golgi network and this prevents premature proteolytic activity [12]. Similarly, cathepsins are synthesized as inactive zymogens and need to be proteolytically activated [11]. TMPRSS2 undergoes autoproteolytic cleavage and this can lead to the secretion of soluble TMPRSS [13,14], but the intracellular location of this cleavage remains ill-defined. Therefore, not only direct inhibitors of furin and TMPRSS2, but also of proteases mediating their activation and trafficking might block viral invasion. The broad inhibition of lysosomal-Golgi trafficking by apilimod [3,4] might interfere with this proteolytic activation, as indicated by the accumulation of inactive pro-forms of cathepsin A and D in apilimod-treated cell lines [15]. This interference in zymogen activation could thus explain how apilimod might inhibit the activity of plasma membrane-localized proteases [6] and could well underlie its anti-viral activity.

## 4. Does Apilimod Disturb the Immune Response Against SARS-CoV-2?

However, drugs targeting lysosomal proteases will have counter-effective side effects, as particularly the immune system heavily relies on many different proteases. First, antigen presenting cells (APCs) rely on proteases for the processing of antigens for presentation to T cells [16] and apilimod blocks this proteolytic degradation of ingested antigens in cultured macrophages [17,18]. Second, cathepsins are needed for removal of the chaperone Ii that blocks the antigen loading groove of MHC class II [11] and apilimod also blocks this proteolytic cleavage [6]. Third, the activity of PIKfyve is required for trafficking of MHC class II to the cell surface, as we showed that apilimod inhibited this process in cultured dendritic cells [6]. As a consequence of these effects, apilimod strongly reduced the presentation of peptides from influenza in human MHC class II (Figure 1B) [6]. Fourth, apilimod might interfere with innate antiviral responses, as it was found to induce expression of activating transcription factor 3 (ATF3) in cultured plasmacytoid dendritic cells, which in turn represses production of anti-viral type I interferons [19]. Apilimod can thus be expected to dampen the immune response against SARS-CoV-2.

The dampening of T cell responses by apilimod might be especially detrimental in COVID-19 patients, since apilimod can be expected to aggravate the already impaired T cell immunity observed in these patients (Figure 1B) [20,21,22]. COVID-19 patients often suffer from lymphocytopenia [16]. A recent profiling of immune cells from blood of COVID-19 patients revealed a reduced expression of MHC class II and lower production of pro-inflammatory cytokines compared to healthy controls (Figure 1B) [16,20]. SARS-CoV-2 infection blocks expression of type I interferons (Figure 1B) by myeloid [16] and other cells [23,24,25] and lower levels of these cytokines are detected in serum of SARS-CoV-2 patients [26,27].

We therefore warn that apilimod and other drugs that target proteases may further suppress the immune system in COVID-19 patients and additional caution has to be applied in clinical trials.

## Figures and Tables

**Figure 1 cells-10-00030-f001:**
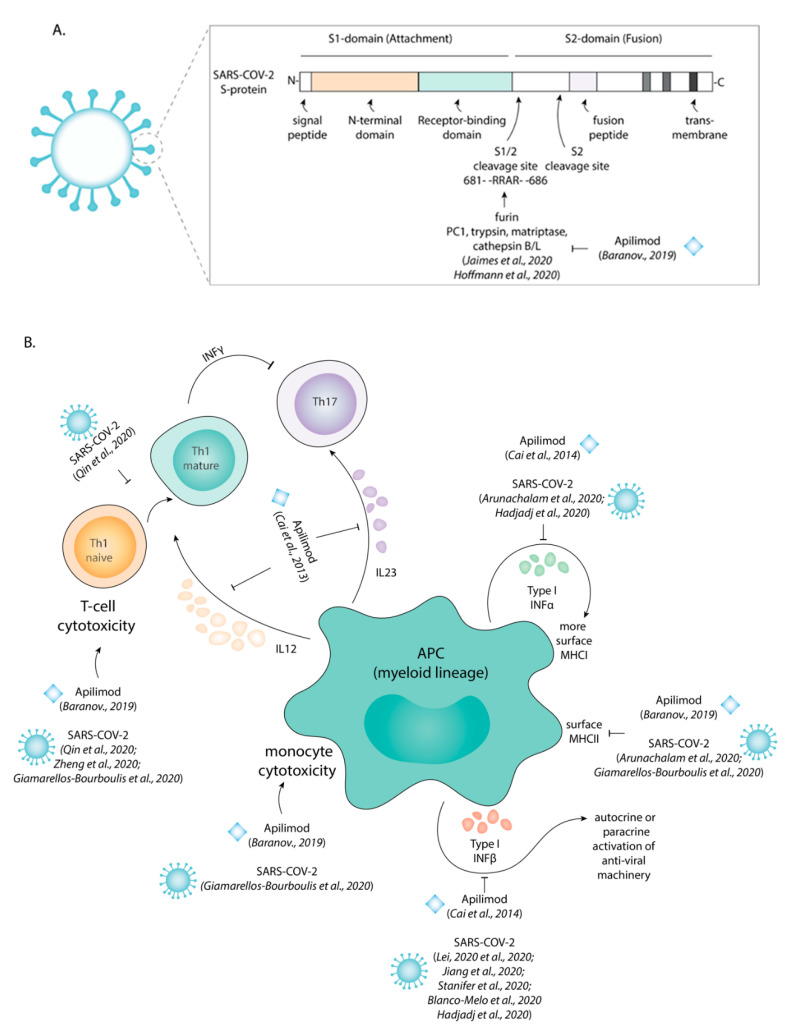
SARS-CoV-2 and apilimod both inhibit the immune system in a similar manner. (**A**) Scheme of the viral S protein indicating the functional domains and the two proteolytic activation sites S1/2 and S2′. Apilimod, an inhibitor of PIKfyve, interferes with the endo/lysosomal trafficking and can indirectly block the activation of proteases as shown for Cathepsin B and L. Apilimod thereby likely interferes with proteolytic activation of the S protein and prevents host cell invasion. (**B**) Both upon infection with SARS-CoV-2 and upon exposure to apilimod, antigen presenting cells (APC) express lower levels of surface MHC class II (HLA-DR; MHCII) and produce less type I interferons (INF-α/β).

## Data Availability

No new data were created or analyzed in this study. Data sharing is not applicable to this article.

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
