# Peer review of "The PIKfyve Inhibitor Apilimod: A Double-Edged Sword against COVID-19"

_cells, 2020, doi:10.3390/cells10010030_

Round 1

Reviewer 1 Report

The authors discuss the pros and cons of using Apilimod as a treatment for Sars-cov-2. Apilimod potently inhibits viral replication in iPSC-derived pneumocyte-like cells, but it also profoundly impairs lysosome to endosome and to TGN membrane trafficking. The authors speculate on the possible  mechanisms linking the impaired endocytic trafficking to the antiviral effect of Apilimod.

The authors hypotesize that apilimod might indirectly inhibit the Tmprss2 activity by impairing its activation by proteolytic cleavage assuming that this occurs in the endocytic or late Golgi compartments. In fact the analogy the authors propose between the activation of furin by proteolytic activation in the trans-Golgi network and in post-Golgi compartments of endo/lysosomal nature or  of cathepsin and that of  Tmprss2 seems to be not properly taken. In fact the cleavage of Tmprss2 has been reported to occur via autoproteolytic cleavage and to  lead to the secretion of soluble Tmprss2. In addition the intracellular location of the autoproteolytic cleavage remains ill defined. The authors should take into consideration these differences between furin/cathepsin and Tmprss2 and possibly refine their hypothesis.

The discussion on potential detrimental effects of Apilimod on the anti-sars-cov-2 immune response is accurate and the concerns raised by the authors are fully shareable.

Author Response

We thank the reviewer for the positive evaluation of our manuscript and helpful comments. As requested, we now include a description of the autoproteolytic cleavage of TMPRSS2 in our revised manuscript (page 1, marked in yellow).

Reviewer 2 Report

The opinion manuscript of Dr. Baranov et al. addresses the potential effect of apilimod, a PIKFyve inhibitor, against COVID-19. The authors elegantly show the potential effect of apilimod in preventing the cellular invasion of SARS-CoV-2, but also the potential side effects disturbing the immune response against SARS-CoV-2. Therefore, both effects must be taken into account when assessing the clinical benefit of this drug in the clinical trials that are currently being carried out.

In figure 1A, it must be specified that the effect of apilimoid is demonstrated on cathepsin B / L. The legend in figure 1 does not correspond to panel A. The legend must be rewritten to address the content of the two panels.

Author Response

We thank the reviewer for the positive evaluation of our manuscript and helpful comments. We have revised our manuscript as requested, and now clearly specify the inhibition of cathepsin B/L in the figure and mention this in the figure legend. We have also extended the figure legend to better describe the contents of panel A.